# The impact of short-term machine perfusion on the risk of cancer recurrence after rat liver transplantation with donors after circulatory death

**Graziano Oldani**[1,2]*, **Andrea Peloso**[1,2], **Florence Slits**[3], **Quentin Gex**[3], **Vaihere Delaune**[1,2], **Lorenzo A. Orci**[1,2], **Yohan van de Looij**[4,5], **Didier J. Colin**[6], **Stéphane Germain**[6], **Claudio de Vito**[2,7], **Laura Rubbia-Brandt**[2,7], **Stéphanie Lacotte**[3©], **Christian Toso**[1,2©]

1 Division of Abdominal Surgery, Department of Surgery, Faculty of Medicine, University of Geneva, Geneva, Switzerland, 2 Hepato-Pancreato-Biliary Centre, Geneva University Hospitals, Geneva, Switzerland, 3 Division of Abdominal Surgery, Department of Surgery, Faculty of Medicine, University of Geneva, Geneva, Switzerland, 4 Division of Child Development & Growth, University Children's Hospital Geneva, Geneva, Switzerland, 5 Institute of Translational Molecular Imaging, University of Geneva, Geneva, Switzerland, 6 MicroPET/SPECT/CT Imaging Laboratory, Centre for BioMedical Imaging, Geneva University Hospitals and University of Geneva, Geneva, Switzerland, 7 Division of Clinical Pathology, Department of Pathology and Immunology, Geneva University Hospitals and University of Geneva, Geneva, Switzerland

© These authors contributed equally to this work.
* graziano.oldani@hcuge.ch

**Data Availability Statement:** All relevant data are within the paper and its Supporting Information files.

## Abstract

Hypothermic and normothermic *ex vivo* liver perfusions promote organ recovery after donation after circulatory death (DCD). We tested whether these perfusions can reduce the risk of hepatocellular carcinoma (HCC) recurrence in a 1h-DCD syngeneic transplantation model, using Fischer F344 rats. DCD grafts were machine perfused for 2h with hypothermic perfusion (HOPE) or normothermic perfusion (NORMO), and transplanted. After reperfusion, we injected HCC cells into the vena porta. On day 28 after transplantation, we assessed tumour volumes by MRI. Control rats included transplantations with Fresh and non-perfused DCD livers. We observed apoptotic-necrotic hepatocyte foci in all DCD grafts, which were more visible than in the Fresh liver grafts. Normothermic perfusion allowed a faster post-transplant recovery, with lower day 1 levels of transaminases compared with the other DCD. Overall, survival was similar in all four groups and all animals developed HCCs. Total tumor volume was lower in the Fresh liver recipients compared to the DCD and DCD +HOPE recipients. Volumes in DCD+NORMO recipients were not significantly different from those in the Fresh group. This experiment confirms that ischemia/reperfusion injury promotes HCC cell engraftment/growth after DCD liver transplantation. Using the present extreme 1h ischemia model, both hypothermic and normothermic perfusions were not effective in reducing this risk.

**Funding:** This study was supported by grants from the Swiss National Science Foundation (PP00P3_139021), the Geneva Cancer League (ref 1509), the Minkoff Foundation, the Artères Foundation, and the Insuleman Foundation. The funders had no role in study design, data collection and analysis, decision to publish, or preparation of the manuscript.

**Competing interests:** The authors have declared that no competing interests exist.

**Abbreviations:** HCC, hepatocellular carcinoma; DCD, donation after circulatory death; HOPE, hypothermic oxygenated perfusion; NORMO, normothermic blood-based perfusion; DBD, donation after brain death; POD, post operatory day; FOV, field of view; ET, echo time; RT, repetition time; MRI, magnetic resonance imaging; PCR, polymerase chain reaction; HMGB1, high mobility group box 1; Hif1a, hypoxia-inducible factor 1; Rplp1, ribosomal protein large P1; Serpine 1, plasminogen activator inhibitor type 1; Hmox1, heme oxygenase 1; Irp94, ischemia-responsive protein 94; I/R, ischemia/reperfusion; AST, aspartate aminotransferase; ALT, alanine aminotransferase; TTV, total tumor volume.

## Introduction

Over the recent years, transplantation of livers retrieved after donation after circulatory death (DCD) has increased significantly in most parts of the world. Despite being associated to more ischemia/reperfusion lesions, and higher risks of primary non/poor function and ischemic cholangiopathy, such transplantations lead to similar post-transplant survivals as those achieved after donation after brain death (DBD)[1–5].

A point of attention is the clear experimental association between ischemia/reperfusion lesions and hepatocellular carcinoma (HCC) cell engraftment and growth [6–8]. Although still a matter of debate, clinical data suggest that ischemia/reperfusion is also associated with a higher risk of post-transplant cancer recurrence[8–11]. Overall, the more marginal the donor, the more ischemia/reperfusion lesion, and the higher the risk of recurrence, as illustrated by DCD donors with prolonged primary warm ischemia times carrying higher HCC recurrence rates compared to leaner counterparts [8].

In order to minimize such lesions, current DCD transplantation practices include the use of *in situ* (normothermic regional perfusion) and *ex vivo* (hypothermic oxygenated or normothermic machine perfusion) organ conditioning. These techniques can rescue injured organs and improve transplant outcomes [12, 13] [2–5, 14].

We previously reported that an *in situ* normothermic liver reperfusion can both reverse ischemia/reperfusion lesions, and the growth and engraftment of HCC cells[6]. Going one step further, the current study explores the potential of hypothermic and normothermic *ex vivo* machine perfusion to reduce HCC recurrence using a rat DCD liver transplantation model.

## Materials and methods

### Animals

Female Fischer (F344) rats (Janvier Labs, Le Genest-Saint-Isle, France) were used for all experiments. Donors weighed 176 [155–193]g and recipients weighed 191 [170–220]g. They were cared for according to the international guidelines on animal care, and ethical approval was obtained from the ethical committee at the University of Geneva and from the Geneva veterinary authorities (GE53/17 and GE178/17). Rats were housed in open cages, with an enriched environment (Aspen Tapvei, Harjumaa, Estonia), under 12/12-h light/dark cycles and free access to water and standard chow (RM3 diet by Special Diet Services, Essex, UK). During the first four hours after surgery, animals were kept in standard individual cages, with the same enrichment, and having free access to water, and food. No formal randomization was applied when forming the study groups.

### Study design

We performed syngeneic arterialized orthotopic liver transplantations using 1h-DCD grafts. Prior to implantation, organs were treated *ex vivo* with either hypothermic oxygenated perfusion (HOPE) or normothermic blood-based perfusion (NORMO). Right after reperfusion, Fisher rat-derived JM-1[6] HCC cells were injected into the portal stream. Tumor growth was assessed by *ex vivo* MRI 28 days after transplantation. Controls were rats transplanted with fresh livers or 1h-DCD livers non-*ex vivo* perfused (Fig 1). Seven recipients were used in each group.

### DCD liver graft procurement

Rat liver procurement was performed under isoflurane anaesthesia (2.5–3.5%) and buprenorphine analgesia (0.1mg/kg subcutaneous given at induction). Liver isolation steps in the donor

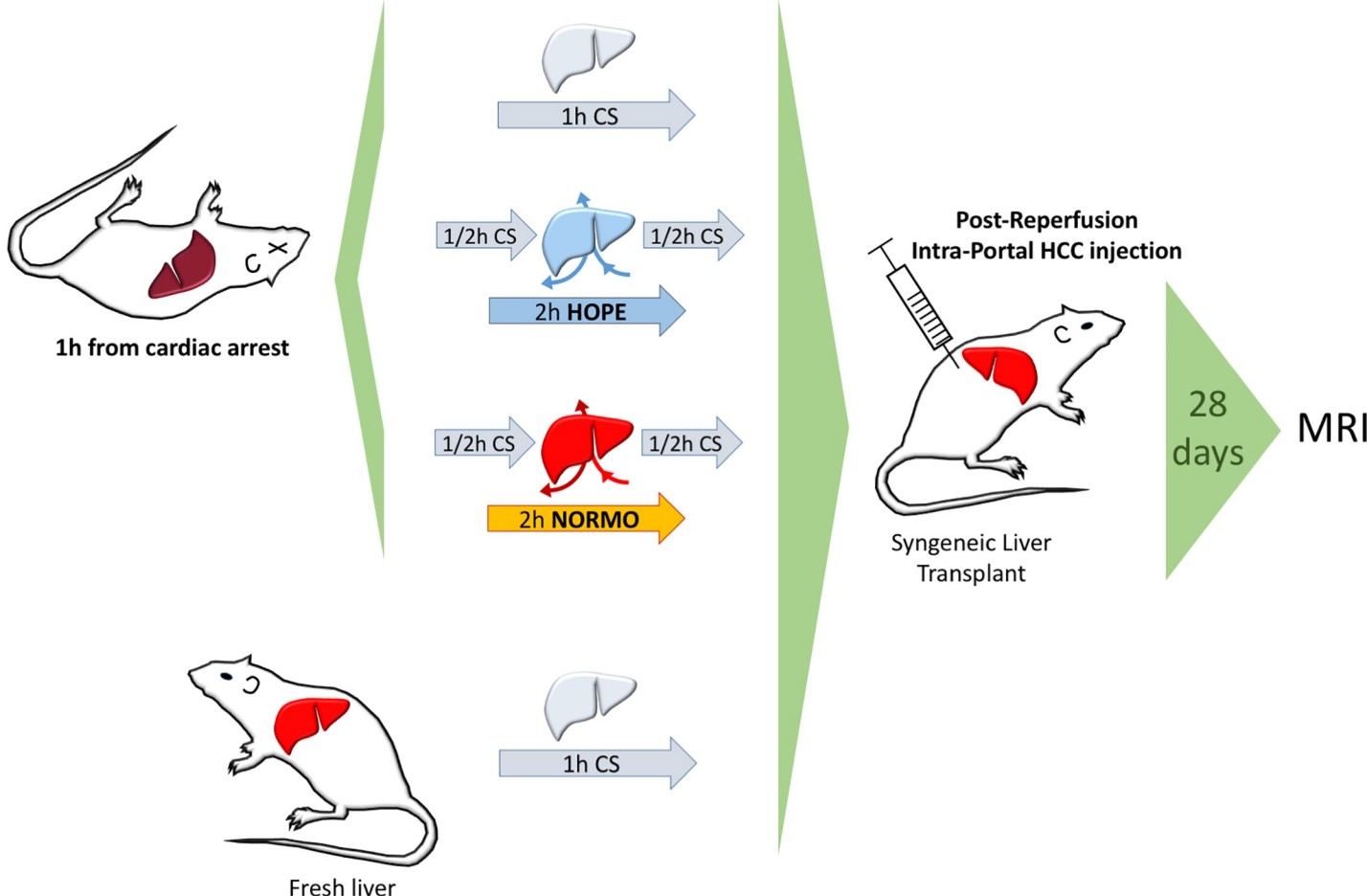

**Fig 1. Experimental design.** CS = cold storage, HOPE = hypothermic oxygenated perfusion, NORMO = normothermic blood-based perfusion, HCC = hepatocellular carcinoma, MRI = magnetic resonance imaging.

were performed as previously described [15]. In addition, the proper hepatic artery was carefully dissected and the gastroduodenal artery divided between 7–0 silk ties. The common hepatic artery was entirely dissected and ligated at its origin. A short fused silica stent (360 µm×150µm, IDEX, Oak Harbor, USA) was then inserted into its lumen and secured with a 7–0 silk tie (as previously described [16]).

Once the liver was completely isolated, the diaphragm was transected on each side. 1h-DCD time was measured from the onset of ventricular fibrillation, as previously described[17]. The liver was then flushed through the portal vein with 20mL cold IGL-1 solution (Institut Georges Lopez, Lyon, France), carrying 100UI of heparin.

All liver grafts had a total of 1h static cold storage in IGL-1 solution. The transitions from static conservation to *ex vivo* perfusions followed a strict timeline, as illustrated in Table 1.

### *Ex Vivo* Hypothermic Oxygenated perfusion (HOPE)

Grafts were perfused on a Radnoti liver/kidney system (mod. 130003. Radnoti glass technology inc, Monrovia, CA, USA). Hypothermic perfusion was performed for 2 hours, applying the most commonly used parameters[12, 17](Table 2). We chose to use non-complemented IGL-1 solution in order to match the 'HOPE for Human Liver Graft' clinical trial (NCT01317342. PI: Dr P. Dutowski, University Hospital of Zurich, Switzerland).

**Table 1. Graft cold storage-perfusion scheme.**

| Group | Donor flush | Static cold storage | | N. Saline flush duration | Ex vivo perfusion | | Static cold storage | | N. Saline cold flush duration |
|---|---|---|---|---|---|---|---|---|---|
| | | Solution | Duration | | Solution | Duration | Solution | Duration | |
| Fresh | IGL-1 | IGL-1 Backtable | 1h | n/a | n/a | | n/a | | 1min |
| DCD | IGL-1 | IGL-1 Backtable | 1h | n/a | n/a | | n/a | | 1min |
| DCD+ HOPE | IGL-1 | IGL-1 Connection to perfusion chamber | 0.5h | n/a | IGL-1 | 2h | IGL-1 Backtable | 0.5h | 1min |
| DCD+ NORMO | IGL-1 | IGL-1 Connection to perfusion chamber | 0.5h | 1 min | ½ Blood ½ N. Saline | 2h | IGL-1 Backtable | 0.5h | 1min |

## *Ex Vivo* Normothermic perfusion (NORMO)

Normothermic perfusion was performed for 2h under pressure control, with the same perfusion apparatus, chamber and circuit configurations used for hypothermic perfusion (S1 Fig, Table 2). Grafts were perfused with adult male Fischer rat blood, diluted 1:1 with normal saline (total volume 50mL, haematocrit 20–22%). 150 UI of heparin were added to the solution in order to prevent clot formation. The perfusate was sampled for pH every 30 minutes. Bile was drained out the perfusion chamber through a fine silicon tubing and the production was recorded. Inflow and outflow haemoglobin saturation was measured with a saturometer (GO2 Achieve, NONIN, Plymouth, MN, USA) through the tubing, and graft oxygen uptake was calculated using the formula: $\Delta$ *Saturation* $\times$ *Hemoglobin* $\times \frac{perfusion\ flow}{liver\ mass} \times 1.34$ [18].

## Liver transplantation

Recipient operation was performed under isoflurane anesthesia (2.5–0.5%) and buprenorphine analgesia (0.02mg/kg subcutaneous given at induction). We used the standard technique developed in our lab[6, 15], with the addition of a stent-assisted arterial reconstruction. This anastomosis was performed after portal reperfusion and IVC reconstruction, by connecting the donor and the recipient's common hepatic arteries on a fused silica stent, as previously described[16]. Of note, all the recipients underwent spleno-systemic shunting by spleen subcutaneous transposition[19] 3 weeks prior to liver transplant. Fischer rats being particularly susceptible to portal clamping, this strategy allows to obtain high survival rates despite the use of damaged grafts[6]. Spleno-systemic shunting in fact reduces splanchnic congestion and the consequent release of vasoplegic molecules responsible for shock after portal unclamping[20].

After surgery, recipient rats were kept in heated single cages for 12h with free access to food and water. Analgesia (subcutaneous buprenorphine 0.005mg/kg) was administered every 8h during the first 48h.

**Table 2. *Ex vivo* perfusions parameters.**

| | HOPE | NORMO |
|---|---|---|
| Solution | IGL-1 (50mL) Heparin (150UI) | ½ blood ½ normal saline (50mL) Heparin (150UI) |
| Temperature | 4˚C | 37˚C |
| $0_2$ pressure | 350mmHg | 350mmHg |
| Haemoglobin saturation | na | 99% |
| Hydrostatic Pressure | 4 mmHg | 8 mmHg |
| Flow | Pressure dependent | Pressure dependent |

## HCC cells injection

JM-1 cells[21] kindly provided by Dr G. Michalopoulos (Duke University Medical Center, Durham, NC), were maintained in Dulbecco's modified Eagle medium (Gibco, Paisley, UK) in standard culture conditions. Twenty minutes after portal reperfusion, $5x10^5$ JM-1 cells suspended in 600µl NaCl 0.9% were injected into the portal vein over 30 seconds, using a 25 G needle. The hole left by the needle was closed with a single 9–0 nylon 'figure of eight' suture.

## Magnetic Resonance Imaging (MRI)-based HCC volume assessment

Transplanted rats were sacrificed on post operatory day (POD) 28 by exsanguination under general anaesthesia. Explanted liver grafts were perfused and stored with formalin. Scans were performed on a 3.0T MR scanner (Nanoscan 3T, RS2D, Mundolsheim, France) with a birdcage coil of 3.5cm diameter. After automatic adjustment of the B0 homogeneity, T1w and T2w images were acquired. For T1w, a gradient echo 3D sequence was used with following parameters: field of view (FOV) 60x40 mm, matrix size 192x128, echo time (ET)/repetition time (RT) 3.8/50ms, flip angle 50˚, 56 slices x 0.5mm thickness and 10 averages. For T2w images, a fast Spin Echo sequence was used with following parameters: FOV 60x40 mm, matrix size 192x128, ET/RT 87.5/4625ms, 35 slices x 0.8 mm thickness and 8 averages.

MRI images were loaded on OsiriX DICOM Viewer (Pixmeo, Geneva, Switzerland) and tumours' contour was manually marked on each slide. Volumes were then calculated using the built-in application.

## Real-Time Polymerase Chain Reaction (PCR)

qPCR assessments were performed on pre-implantation liver graft parenchyma. Total RNA was prepared and purified from frozen-crushed liver powder using the RNeasy Mini Kit (Qiagen, Germantown, MD), and according to the manufacturer's instructions. cDNA was synthesized from one µg of RNA by extending a mix of random primers with the High Capacity cDNA Reverse Transcription Kit in the presence of RNase Inhibitor (Applied Biosystems, Carlsbad, CA). The relative quantity of each transcript was normalized according to the expression of rplp1 (ribosomal protein large P1). Primers for HMGB1 (high mobility group box 1, NM_012963) and Hif1a (hypoxia-inducible factor 1, NM_024359.1) were bought from Qiagen. The other primers were homemade-designed and the sequences are available upon request. The following primers were used: rplp1 (ribosomal protein large P1, NM_001007604), Serpine1 (plasminogen activator inhibitor type 1, NM_012620.1), Hmox1 (heme oxygenase 1, NM_012580.2), irp94 (ischemia-responsive protein 94 (also known as Hsp4), AF_077354), and F3 (Tissue factor, NM_013057.2). Amplification reactions were performed in a total volume of 20µl using either a Thermocycler sequence detector (BioRad CFX96) with Takyon for SYBR Assay—No ROX (Eurogentec, Seraing, Belgium).

## Transaminase and HMGB1 Serum Quantification

24h post-transplant aspartate aminotransferase (AST), alanine aminotransferase (ALT), levels were measured on serum from centrifuged blood (11,000 rpm for 7 minutes at 4 ˚C) using a multiple strip machine (RefloVet Plus, Roche, Basel, Switzerland). HMGB1 ELISA assay was performed according to manufacturers' instructions (IBL International GMBH).

## Light microscopy

Liver sections were stained with hematoxylin and eosin (Sigma, Buchs, Switzerland). Histological studies were performed blindly by an expert in liver anatomo-pathology.

## Statistical analysis

Data were expressed as median [range] and were analysed using Prism 7 (GraphPad Software Inc., CA, USA). Survivals were plotted according to Kaplan-Meier and compared using Log-rank (Mantel-Cox) test. All other continuous data were compared using Mann-Whitney test. The limit for statistical significance was set at $p = 0.05$.

## Results

### DCD donors

After cutting the diaphragm, respiratory arrest occurred within 7 [4–11]min in the DCD group, 6 [4–10]min in the DCD+HOPE, and 7 [3–12]min in the DCD+NORMO (all p>0.05 in pairing assessments). Cardiac arrest occurred after 9 [6–13]min, 9 [6–12]min and 10 [6–13] min respectively (all p>0.05).

### Hypothermic perfusion of DCD liver grafts

**Perfusion flow and resistance.**   The early HOPE perfusion flow was low (0.22 [0.19–0.23] mL/min/g). It increased during the first 30 minutes and reached a plateau after 1 hour (0.29 [0.26–0.33] mL/min/g; Fig 2A left). The calculated resistance followed the opposite trend, dropping from 2.40 [2.28–2.71] mmHg/(mL/min) to 1.72 [1.58–2.00] mmHg/(mL/min) (Fig 2A right) over the first 30 minutes of perfusion.

**Oxygen uptake and bile production.**   Inflow and outflow oxygen concentrations were similar in the perfusion solution (inflow 39.15 [38.95–39.27] %, outflow 39.05 [38.83–39.18] %, p>0.5), suggesting no oxygen uptake (or lower than the instrument sensitivity). No bile production was detected under hypothermic perfusion.

### Normothermic perfusion of DCD liver grafts

**Perfusion flow and resistance.**   The early NORMO perfusion flow was low (0.22 [0.29–0.48] mL/min/g). It rapidly increased during the first 30 minutes and reached a plateau after 1 hour (1.10 [0.67–1.29] mL/min/g; Fig 2B left). As expected, the calculated resistance showed the opposite trend, dropping from 2.45 [2.18–2.73] mmHg/(mL/min) to 1.07 [0.97–1.26] mmHg/(mL/min)(Fig 2B right) over the first 30 minutes of normothermic perfusion. As shown in Fig 2C the initial resistance to flow was equal in all DCD grafts. However, the normothermic perfusion allowed for a more consistent reduction in perfusion resistance (-63%) compared to the hypothermic approach (-28.33%).

**Oxygen uptake and bile production.**   The early calculated oxygen uptake of the DCD liver grafts was low (0.0016 [0.0013–0.0025] mL/min/g). It rapidly increased to reach a plateau (0.018 [0.0015–0.019] mL/min/g) after 30 minutes (Fig 2D). Bile production followed the same pattern, being 12.6 [6.1–20.2] μL/g over the first 30 minutes, then constantly around 20 μL/g every following half hour (Fig 2D).

**Ischemia/reperfusion injury assessment.**   DCD grafts appeared markedly congested and firm. After the initial cold flushing, some areas remained only partially washed out (red large patches in Fig 3A, right). This phenomenon was possibly related to a local vasospasm and perivascular oedema.

During the hypothermic perfusion the large dark areas were effectively washed out (Fig 3B, comparison between the 1st and 2nd pictures from the left). However, the grafts never recovered fully, as demonstrated by the inhomogeneous colour of the liver parenchyma after the final flush (Fig 3B, 4th column from the left). Further, we still observed macroscopic perfusion abnormality 20 minutes after reperfusion (Fig 3B, last two pictures).

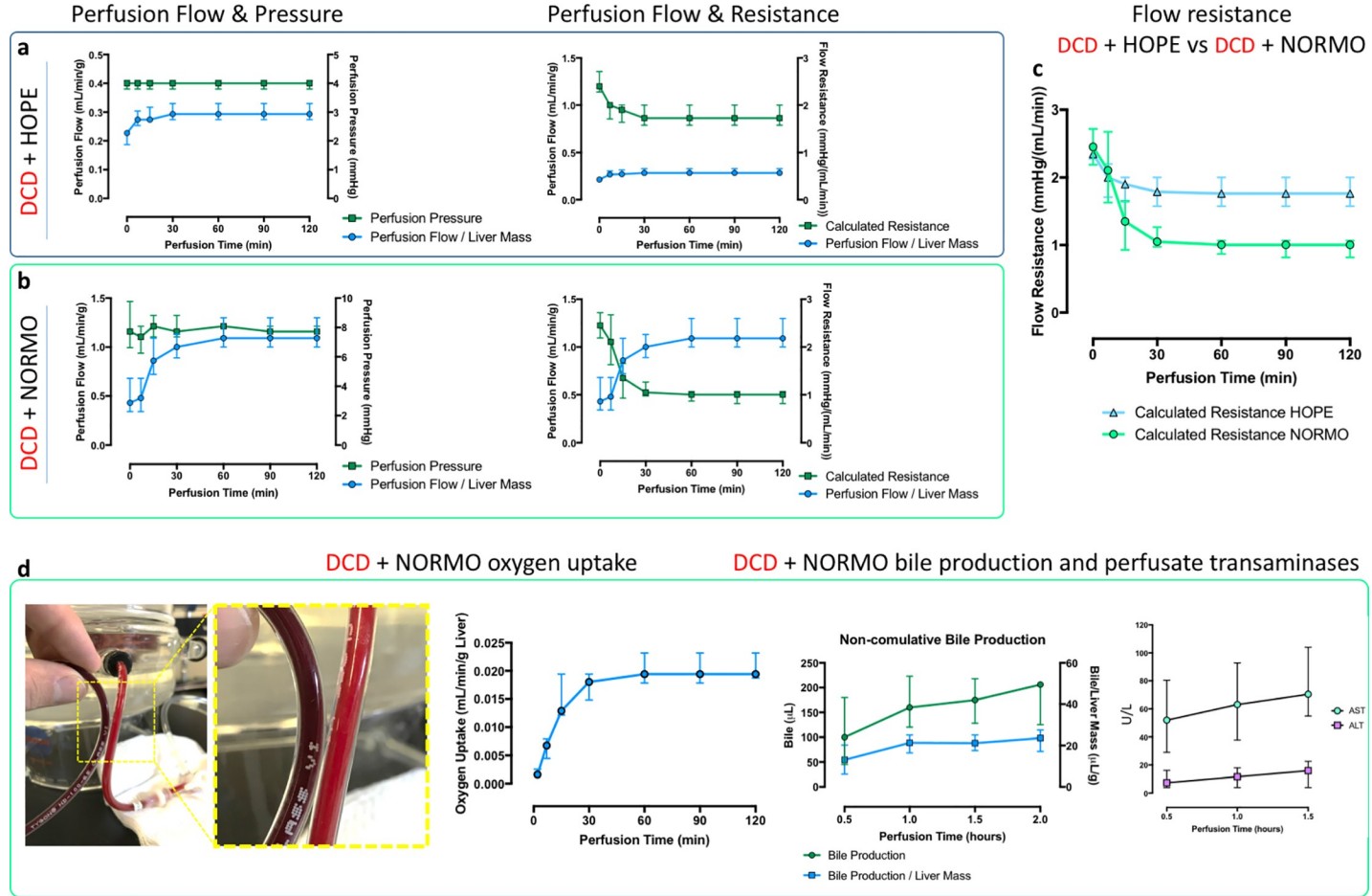

**Fig 2. *Ex vivo* perfused liver graft fluidodynamics and metabolism.** a. Left, hypothermic perfusion of DCD grafts shows modest flow increase over time, under controlled pressure conditions. Right, hypothermic perfusion of DCD grafts shows modest resistance to flow reduction over time. b. Left, normothermic perfusion of DCD grafts shows significant flow increase over time, under controlled pressure conditions. Right, normothermic perfusion of DCD grafts shows significant resistance to flow reduction over time. c. Direct comparison of resistance to flow changes between hypothermic and normothermic *ex vivo* perfusion. d. Left, visual comparison of inflow vs. outflow haemoglobin saturation during normothermic perfusion. Second from left, oxygen uptake of DCD grafts on normothermic perfusion. It shows increasing oxygen consumption that reaches a plateau after one hour from perfusion initiation. Third from left, non-cumulative bile production of DCD grafts on normothermic perfusion. It shows constant rate bile production after the first hour of perfusion. Right, transaminase in the perfusion solution. It shows enzyme accumulation during perfusion, symptom of graft ischemia/reperfusion injury. DCD = donation after circulatory death, HOPE = hypothermic oxygenated perfusion, NORMO = normothermic blood-based perfusion, AST = aspartate aminotransferase, ALT alanine aminotransferase.

During normothermic perfusion, concomitantly to the drop of vascular resistance, most of the hypo-perfused areas disappeared (Fig 3C, 2nd picture from the left), leaving almost no mark after the final flush (Fig 3C, 3rd picture from the left). Only small peripheral areas did not recover, and were still visible 20 minutes after reperfusion (Fig 3C, last two pictures).

Pre-implantation biopsies showed decreased congestion in the DCD-HOPE and the DCD-NORMO, compared to the non-perfused DCD grafts. By contrast all DCD grafts had similar increased number of apoptotic hepatocytes (black arrows in Fig 3C) and a loss of inter-cellular cohesion (compared with the Fresh grafts).

Evidence indicates that Irp94, Hmox1, Hif1a, Serpine1, HMGB1 and F3 genes expression is up-regulated in case of liver hypoxia and ischemic injury[22–25]. The mRNA levels of Serpine 1, F3 and Hmox were increased in the DCD grafts after normothermic perfusion, while there was no difference in the expression of these genes in all other groups (Fig 4A, left column.

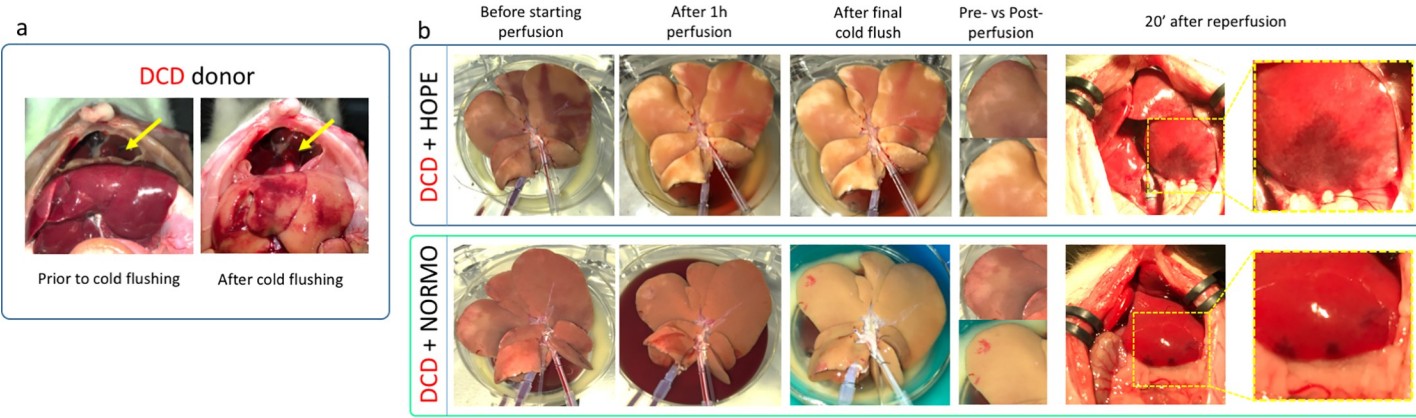

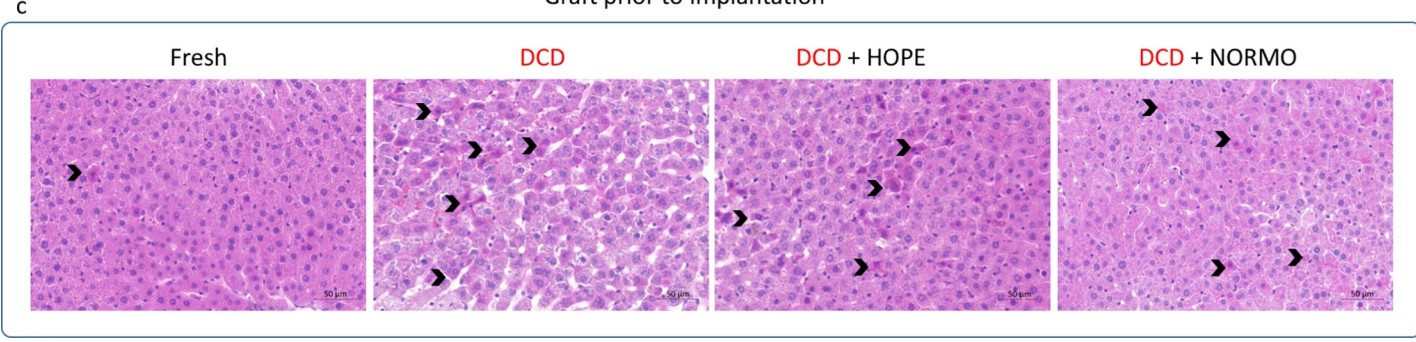

**Fig 3. Ischemia/reperfusion injury in DCD *ex vivo* perfused liver grafts.** a. 1h DCD donor prior and cold flushing. Arrows indicating the wide cut in the diaphragm.b. Comparison of DCD grafts during hypothermic and normothermic perfusion and after reperfusion in the recipient. Normothermic treatment allowed for more homogeneous perfusion of the grafts and reduction of the local vasospasm. c. Histological comparison of liver grafts prior to implantation. All DCD livers presented similar amount of apoptotic hepatocytes (black arrows). DCD and DCD+HOPE grafts presented loss of intercellular cohesion. DCD+NORMO grafts showed loss of intercellular cohesion to a minor extent compared to DCD and DCD+NORMO grafts. DCD = donation after circulatory death, HOPE = hypothermic oxygenated perfusion, NORMO = normothermic blood-based perfusion.

Fresh vs. NORMO p<0.005, DCD vs. NORMO p<0.005, HOPE vs. NORMO p<0.005). Irp94 and Hif1a expression was up-regulated both in the DCD+HOPE and DCD+NORMO, although to a greater extent in the latter group (Fig 4A, right column. DCD+HOPE vs. DCD +NORMO p<0.05). No difference was found in the expression of HMGB1 among all pre-implantation grafts.

24 hours after transplantation Fresh liver recipients showed normal plasma levels of ALT and AST. DCD and DCD+HOPE recipients had very high levels of both AST (Fresh vs. DCD p = 0.012, Fresh vs. DCD+HOPE p = 0.016) and ALT (Fresh vs. DCD p = 0.006, Fresh vs. DCD+HOPE p = 0.008). DCD+NORMO recipients by contrast showed lower levels of both liver enzymes (DCD+HOPE vs. DCD+NORMO p = 0.016 for both AST and ALT, Fig 4B).

The nuclear factor HMGB1 is an early mediator of injury and inflammation after liver ischemia-reperfusion[26]. Serum levels of HMGB1 were lower in the Fresh liver recipients 24 hours after transplantation, while no difference was found among the DCD recipients, whether *ex-vivo* perfused or not (Fresh vs. DCD, DCD+HOPE, DCD+NORMO p<0.05. Fig 4C)

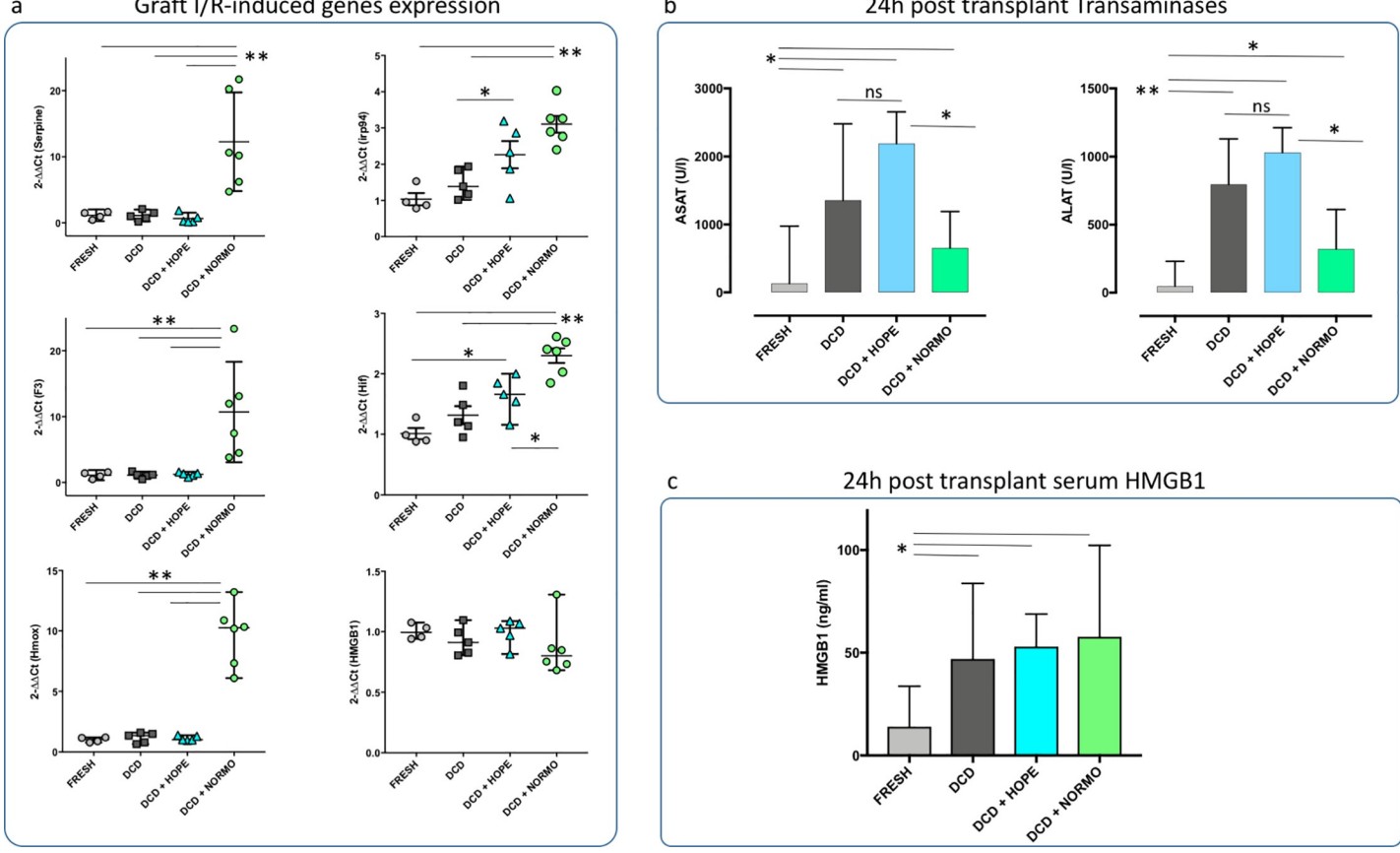

**Fig 4. Pre- and post transplant ischemia/reperfusion injury assessment.** a. Pre implantation graft I/R-induced genes expression (fold increase). The mRNA levels of Serpine 1, F3 and Hmox were increased in the DCD grafts after normothermic perfusion, while there was no difference in the expression of these genes in all other groups (Fresh vs. NORMO p<0.005, DCD vs. NORMO p<0.005, HOPE vs. NORMO p<0.005). Irp94 and Hif1a expression was up-regulated both in the DCD+HOPE and DCD+NORMO, although the latter to a greater extent (DCD+HOPE vs. DCD+NORMO p<0.05). No difference was found in the expression of HMGB1 among all pre implantation grafts. b. 24h post transplant serum transaminase. DCD and DCD+HOPE recipients had very high levels of both AST (Fresh vs. DCD p = 0.012, Fresh vs. DCD+HOPE p = 0.016) and ALT (Fresh vs. DCD p = 0.006, Fresh vs. DCD+HOPE p = 0.008). DCD+NORMO recipients by contrast showed lower levels of both liver enzymes (DCD+HOPE vs. DCD+NORMO p = 0.016 for both AST and ALT). c. 24h post transplant serum HMGB1. Circulating HMGB1 was lower in the Fresh liver recipients, while no difference was found among the DCD recipients, whether *ex-vivo* perfused or not (Fresh vs. DCD, DCD+HOPE, DCD+NORMO p<0.05). HMGB1 = high mobility group box 1, Hif1a = hypoxia-inducible factor 1, rplp1 = ribosomal protein large P1, Serpine1 = plasminogen activator inhibitor type 1, Hmox1 = heme oxygenase 1, irp94 = ischemia-responsive protein 94, I/R = Ischemia/reperfusion, DCD = donation after circulatory death, HOPE = hypothermic oxygenated perfusion, NORMO = normothermic blood-based perfusion, AST = aspartate aminotransferase, ALT alanine aminotransferase.

## Post-Transplant Health, Survival and HCC development

All animals recovered from the transplant procedure. DCD recipients were more prone to bleeding from the liver surface, which was not the case in the DCD+NORMO recipients. None required blood transfusion or volume replacement in the Fresh group. Conversely, two recipients in the DCD and three recipients in the DCD+HOPE group were transfused with 0.5mL of syngeneic full blood, in order to compensate an estimated blood loss of 0.4–0.7 mL.

All DCD recipients were mildly lethargic on the first post-operative day, but DCD +NORMO ones being slightly more active. One animal from the DCD+HOPE and one from the DCD+NORMO groups died on POD 1 (Fig 5A), after having undergone open surgical liver biopsy (see discussion). All the other rats survived till the end of the follow-up (28 days), except for one DCD+NORMO recipient which died of obstructive cholangitis because of one HCC nodule located at the hilum.

All recipients developed HCC nodules. Tumor implant and growth was extremely homogeneous between the Fresh recipients, with a total tumour volume (TTV) range spanning from 0.92 to 2.10 cm$^3$ only (median 1.60 cm$^3$). In contrast, non-*ex-vivo* perfused DCD liver recipients constantly developed larger tumours (TTV: 3.25 [2.49–8.72] cm$^3$, Fresh vs. DCD p = 0.0006), in line with our previous study [6]. DCD+HOPE and DCD+NORMO recipients showed more variability in HCC size. DCD+HOPE tumours were overall larger than those of the Fresh recipients (TTV: 6.44[1.30–16.97]cm$^3$, Fresh vs. DCD+HOPE p = 0.014). DCD +NORMO were not significantly different from the Fresh group (TTV: 1.52[0.08–9.90]cm$^3$, Fresh vs. DCD+NORMO p = 0.62, DCD vs. DCD+NORMO p = 0.23). All recipients developing larger tumours (TTV>4cm$^3$) displayed peritoneal carcinomatosis at time of autopsy. Conversely, none of the recipients developed lung nodules.

## Discussion

This study confirms that the use of marginal, ischemic rat liver grafts increases HCC growth after transplantation. However, hypothermic oxygenated and normothermic *ex vivo* perfusions did not bring a measurable improvement in the post-transplant HCC volumes. This observation is in contrast with our previous observation utilizing liver grafts with a 30-minute warm ischemia (unlike 60 minutes in the present experiments), where we found that *in situ* normothermic reperfusion prior to organ procurement mitigated post-transplant cancer recurrence[6].

Hypothermic and normothermic perfusions proved effective in rescuing liver grafts from ischemic injuries[1, 12–14, 17, 27, 28]. While the former is simpler approach that requires no blood or additives, the latter has the advantage of mimicking physiological conditions, but requires more sophisticated equipment and involves higher costs[29]. Given that there is no consensus on whether one strategy is more effective than the other[30], and none of them was tested in a cancer setting, we challenged both.

Intimal barotrauma has been identified as a major factor in the failure of achieving beneficial *ex vivo* perfusion[12, 31]. Knowing this, we closely monitored the fluid dynamics of the grafts throughout the process and continuously adjusted the perfusion rate in order to maintain low inflow pressures. Therefore, even the minor pressure fluctuations typical of peristaltic pumps were successfully compensated by the use of a semi-open perfusion chamber system.

In DCD liver grafts, intravascular resistance increases due to perivascular swelling [32], the role of small clots being controversial[33]. We observed that hypothermic perfusion could only marginally reduce this phenomenon, its effect being likely generated by the mechanical washout of residual blood in higher resistance areas. Conversely, normothermic perfusion was able to restore almost normal fluid dynamics as early as 1 hour after its initiation. Similarly, liver grafts remained profoundly metabolically hypoactive during the hypothermic perfusion, as demonstrated by the undetectable oxygen uptake, the absence of bile production and no increase in Serpine1, F3 and Hmox expressions. By contrast, normothermic grafts were actually functioning close to their normal physiology, with only small areas of focal necrosis which never recovered during the 2h perfusion.

Unlike previous studies on hypothermic and normothermic perfusions of prolonged-ischemia rat liver grafts[12, 17, 34], we could not appreciate any difference in post-transplant survival. This observation may have multiple explanations. First, we used arterialized grafts. Although it does not affect survival in rodents transplanted with uninjured livers [35, 36], it has been proven to help function recovery in ischemic grafts[37]. Second, spleno-systemic shunts protected the recipients from any portal clamping-related shock, avoiding the synergy bad organ/surgical stress, causing deaths non necessarily related to the status of the grafts only.

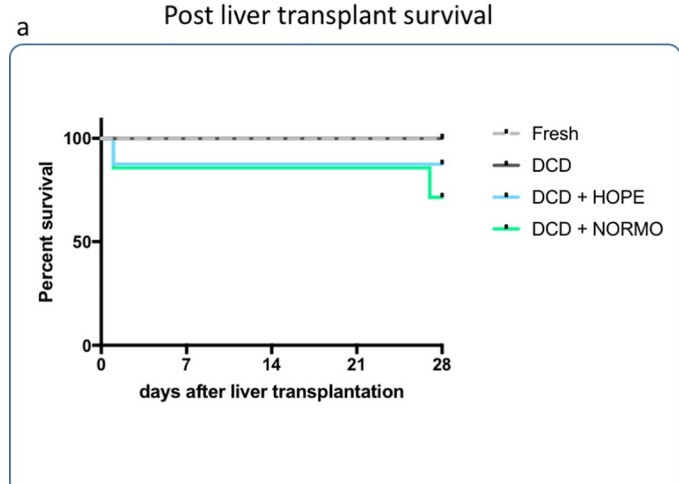
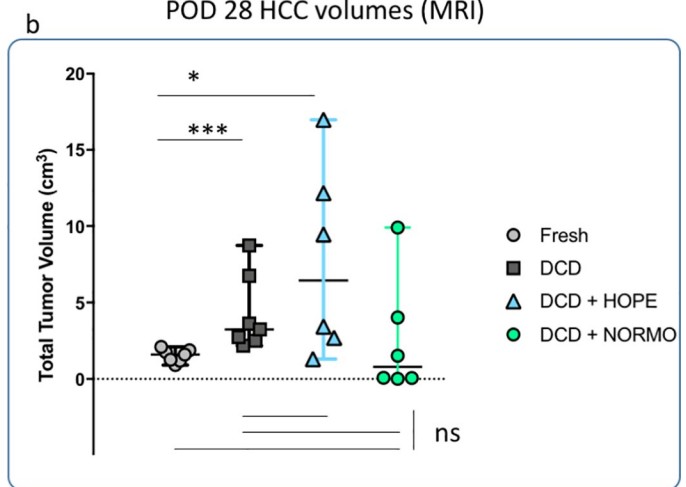

**Fig 5. Post transplant survival and HCC development.** a. Post liver transplant survival. b. POD 28 HCC volumes. Tumour growth was homogeneous between the Fresh recipients. Non *ex-vivo* perfused DCD liver recipients developed larger tumours (Fresh vs. DCD p = 0.0006). DCD+HOPE and DCD+NORMO showed more variability in HCC size. DCD+HOPE tumours were overall larger than those of the Fresh recipients (Fresh vs. DCD+HOPE p = 0.014). DCD+NORMO were not significantly different from the Fresh group (Fresh vs. DCD+NORMO p = 0.62, DCD vs. DCD+NORMO p = 0.23) POD = postoperative day, HCC = hepatocellular carcinoma, DCD = donation after circulatory death, HOPE = hypothermic oxygenated perfusion, NORMO = normothermic blood-based perfusion.

Third, our extensive expertise with rodent liver transplantation[6, 15, 36] helped designing a study with few biases.

Exposing the grafts to longer perfusion times would not have affected the outcomes after hypothermic perfusion, as studies demonstrated a plateauing effect after one hour[1, 2, 12, 13, 17, 27, 31]. In contrast, it takes at least two hours for a normothermic perfusion to have an impact, and we set both NORMO and HOPE protocols at this perfusion time. Moreover, we made this choice to avoid the need for a dialysis circuit or complicated nutrients and electrolytes complementation of the perfusate[28, 34]. This aspect represents a limitation of this study, since longer normothermic perfusions may have led to a better organ recovery[38] and potentially to a decreased HCC implantation/growth.

Another limitation in this study is the absence of a post-reperfusion biopsy in the recipients. It was our original plan to perform a surgical biopsy 24h after implant. However, since most of the recipients in the DCD and DCD+HOPE groups were not fit enough on POD1 to cope with another surgery, we skipped this step to meet local veterinary requirements.

Only two well characterized HCC cell lines have been isolated from inbred rat hepatomas: JM-1 in Fischer and McA-RH7777 in Buffalo rats[39]. Since hepatocarcinoma can be very heterogeneous in humans[40], we aimed at reproducing the current experiments with McA-RH7777 cells. Unfortunately, we found Buffalo rats to have extreme abdominal vascular variations that made the transplantation model unsuitable for optimal standardization. If having failed to use a second cell line may have reduced the translational value of this study, our findings still find support in the behaviour of similar tumors in other species[41]. In particular, our group has demonstrated the increased risk of HCC recurrence in ischemic mouse livers[7, 8].

Overall, the study confirms that ischemic liver grafts are more likely to develop worse HCC recurrence. In the present setting, hypothermic perfusion did not bring a measurable benefit in terms of cancer implantation/growth reduction. Normothermic perfusion helped organ function recovery, but we were not able to demonstrate a consistent effect in reducing cancer

burden. Further studies with shorter ischemic times and longer normothermic perfusion would be of interest.

## Supporting information

**S1 Checklist. NC3Rs ARRIVE guidelines checklist.**
(DOCX)

**S1 Fig. Perfusion chamber and graft connection.** The perfusion chamber is a semi-open system composed of a lower chamber (35x10mm petri dish) glued to the inferior surface of a larger open chamber (petri dish 60x15mm). The two chambers communicate through three 2mm holes in the floor of the upper chamber. Chambers are filled with cold solution and liver graft lays into the upper chamber (the SHVC being previously tied). The vena porta (inflow) and the IVC (outflow) are connected to cannulas fitting their size without excessive stretching. The IVC cannula is connected to a soft silicon tubing linking the upper and the lower chamber. The perfusion solution runs through the graft and is collected into the lower chamber without contaminating the fluid in the upper chamber. The solution collecting into the lower chamber goes to the pump through a single silicon tubing. Of note, the cyclic pressure fluctuations (typical of peristaltic pumps) potentially causing reduced aspiration in the lower chamber are compensated by the holes interconnecting the two chambers. IVC = infrahepatic vena cava.
(TIFF)

## Acknowledgments

We wish to thank Maria-Louisa Izamis and Korkut Uygun for the helpful suggestions.

## Author Contributions

**Conceptualization:** Graziano Oldani, Stéphanie Lacotte.

**Data curation:** Florence Slits, Stéphanie Lacotte.

**Formal analysis:** Stéphanie Lacotte.

**Funding acquisition:** Christian Toso.

**Investigation:** Graziano Oldani, Andrea Peloso, Claudio de Vito, Laura Rubbia-Brandt, Stéphanie Lacotte.

**Methodology:** Graziano Oldani, Andrea Peloso, Florence Slits, Quentin Gex, Vaihere Delaune, Yohan van de Looij, Didier J. Colin, Stéphane Germain, Stéphanie Lacotte.

**Project administration:** Graziano Oldani.

**Software:** Yohan van de Looij.

**Supervision:** Christian Toso.

**Writing – original draft:** Graziano Oldani, Christian Toso.

**Writing – review & editing:** Graziano Oldani, Lorenzo A. Orci, Stéphanie Lacotte.

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
