## [Decision Letter · Decision Letter 0]

1 Aug 2019

PONE-D-19-16594

The impact of short-term machine perfusion on the risk of cancer recurrence after rat liver transplantation with donors after circulatory death

PLOS ONE

Dear Dr Oldani,

Thank you for submitting your manuscript to PLOS ONE. After careful consideration, we feel that it has merit but does not fully meet PLOS ONE’s publication criteria as it currently stands. Therefore, we invite you to submit a revised version of the manuscript that addresses the points raised during the review process.

Specifically, Authors should address issues pertaining heterogeneity of HCC cell lines and how their finding relate to this phenotypic feature.

We would appreciate receiving your revised manuscript by Sep 15 2019 11:59PM. To enhance the reproducibility of your results, we recommend that if applicable you deposit your laboratory protocols in protocols.io, where a protocol can be assigned its own identifier (DOI) such that it can be cited independently in the future. For instructions see: http://journals.plos.org/plosone/s/submission-guidelines#loc-laboratory-protocols

We look forward to receiving your revised manuscript.

Kind regards,

Ezio Laconi, MD, PhD

Academic Editor

PLOS ONE

Journal Requirements:

1. To comply with PLOS ONE submissions requirements, in your Methods section, please provide additional information on the animal research and ensure you have included details on (1) methods of sacrifice, (2) methods of anesthesia and/or analgesia, and (3) efforts to alleviate suffering.

2. As part of your revision, please complete and submit a copy of the ARRIVE Guidelines checklist, a document that aims to improve experimental reporting and reproducibility of animal studies for purposes of post-publication data analysis and reproducibility: https://www.nc3rs.org.uk/arrive-guidelines. Please include your completed checklist as a Supporting Information file. Note that if your paper is accepted for publication, this checklist will be published as part of your article.

3. In your Methods section, please give the sources of any cell lines used in your study.

4. Please note that all PLOS journals ask authors to adhere to our policies for sharing of data and materials: https://journals.plos.org/plosone/s/data-availability. According to PLOS ONE’s Data Availability policy, we require that the minimal dataset underlying results reported in the submission must be made immediately and freely available at the time of publication. As such, please remove any instances of 'unpublished data' or 'data not shown' in your manuscript and replace these with either the relevant data (in the form of additional figures, tables or descriptive text, as appropriate), a citation to where the data can be found, or remove altogether any statements supported by data not presented in the manuscript.

Reviewers' comments:

Reviewer's Responses to Questions

**Comments to the Author**

1. Is the manuscript technically sound, and do the data support the conclusions?

Reviewer #1: Partly

Reviewer #2: Yes

2. Has the statistical analysis been performed appropriately and rigorously? 

Reviewer #1: No

Reviewer #2: Yes

3. Have the authors made all data underlying the findings in their manuscript fully available?

Reviewer #1: Yes

Reviewer #2: Yes

4. Is the manuscript presented in an intelligible fashion and written in standard English?

Reviewer #1: Yes

Reviewer #2: Yes

5. Review Comments to the Author

Reviewer #1: Oldani G. and co authors, in the current paper, examined the impact of hypothermic and normothermic ex vivo liver perfusions on JM-1 neoplastic cells growth in transplanted rat liver. In comparison with fresh transplanted rats, both methods of “ex vivo” perfusion determined increase of HCC growth, thus suggesting a role in ischemia-reperfusion injury in supporting tumor proliferation and spreading.

Comments

1) The results of the study were based on the use of JM-1 line only. Several other HCC cell lines are available and, on the other hand, HCC in human denotes several phenotypic differences between subjects. For all the above, the authors should comment the limits of the extrapolation of these results in the setting of HCC in human liver transplantation.

2) Since three groups were included in the study, ANOVA should be the appropriate test for statistical comparison.

3) The study is mainly observational and putative mechanisms relating ischemia-reperfusion injury to increase tumor proliferation should be at list mentioned in discussion. In my opinion, this line of research would benefit in the future by testing HCC growth and spreading genes in “in vitro” easily adaptable systems.

Reviewer #2: This is a very interesting study and the findings are very robust.

The group performed DDLT using DCD grafts in rats compared with fresh liver grafts. The DCD grafts were divided into two groups using HOPE or NORMO as the machine perfusion. While the findings of transaminases post-machine perfusion was somewhat expected, I find it difficult to understand why there are more peritoneal metastasis in the study, and it was particularly related to the tumor size.

The authors tried to induce tumour growth by injecting the designated tumour cells with HCCs via portal vein after repercussion and monitor for tumor growth. However, how does that translate into the distribution of tumours in different parts of the liver was not explained. More importantly, I am curious to know how these larger tumours were able to develop into peritoneal carcinomatosis within 28 days and the details of this data was not available in the results section.

Overall, it is an interesting study, proving the concept that the ischaemic environment promotes tumour progression but the actual pathological process involved, particularly from the immunological aspects was not clear.

6. PLOS authors have the option to publish the peer review history of their article (what does this mean?). If published, this will include your full peer review and any attached files.

Reviewer #1: No

Reviewer #2: No

---

## [Author Response · Author response to Decision Letter 0]

29 Sep 2019

Reviewer #1: 

1) The results of the study were based on the use of JM-1 line only. Several other HCC cell lines are available and, on the other hand, HCC in human denotes several phenotypic differences between subjects. For all the above, the authors should comment the limits of the extrapolation of these results in the setting of HCC in human liver transplantation.

R1: Thank you for the appropriate comment. We implemented the discussion explaining why we were not able to use other cell lines, which was our original plan. 

2) Since three groups were included in the study, ANOVA should be the appropriate test for statistical comparison.

R2: Thank you for the comment. We partially disagree since our aim was to compare each study group directly with the control groups (being the results of the two control groups, predicted by our previous studies). For completeness, we ran ANOVA tests to follow the suggestion and the results did not change. 

3) The study is mainly observational and putative mechanisms relating ischemia-reperfusion injury to increase tumor proliferation should be at list mentioned in discussion. In my opinion, this line of research would benefit in the future by testing HCC growth and spreading genes in “in vitro” easily adaptable systems.

R3: Thank you for the comment. With this study we did not aim to prove mechanisms, as we already performed more targeted experiments in simpler settings in the past. We implemented the discussion including these citations. 

Reviewer #2: This is a very interesting study and the findings are very robust.

1) The group performed DDLT using DCD grafts in rats compared with fresh liver grafts. The DCD grafts were divided into two groups using HOPE or NORMO as the machine perfusion. While the findings of transaminases post-machine perfusion was somewhat expected, I find it difficult to understand why there are more peritoneal metastasis in the study, and it was particularly related to the tumor size.

R1: Thank you for the comment. We found more peritoneal involvement when tumors were large enough to infiltrate beyond liver capsule. We did not assess whether these tumors had switched to more aggressive phenotypes.

2) The authors tried to induce tumour growth by injecting the designated tumour cells with HCCs via portal vein after repercussion and monitor for tumor growth. However, how does that translate into the distribution of tumours in different parts of the liver was not explained. More importantly, I am curious to know how these larger tumours were able to develop into peritoneal carcinomatosis within 28 days and the details of this data was not available in the results section.

R2: Thank you for the comment. It was very hard to identify a tumor distribution pattern. Rat livers are small, and tumors grow quickly, passing from non being detectable to merging nodules in few days. 

3) Overall, it is an interesting study, proving the concept that the ischaemic environment promotes tumour progression but the actual pathological process involved, particularly from the immunological aspects was not clear.

R3: Thank you for the comment. We have already published (on PLOS) the role of the immune system on cancer recurrence after transplant, using a similar model. It would have been interesting to assess it in this context too (as many other things), but it was not the focus of the study. 

We hope this revision satisfies your expectations.

---

## [Decision Letter · Decision Letter 1]

24 Oct 2019

The impact of short-term machine perfusion on the risk of cancer recurrence after rat liver transplantation with donors after circulatory death

PONE-D-19-16594R1

Dear Dr. Oldani,

We are pleased to inform you that your manuscript has been judged scientifically suitable for publication and will be formally accepted for publication once it complies with all outstanding technical requirements.

With kind regards,

Ezio Laconi, MD, PhD

Academic Editor

PLOS ONE

Additional Editor Comments (optional):

Reviewers' comments:

Reviewer's Responses to Questions

**Comments to the Author**

1. If the authors have adequately addressed your comments raised in a previous round of review and you feel that this manuscript is now acceptable for publication, you may indicate that here to bypass the “Comments to the Author” section, enter your conflict of interest statement in the “Confidential to Editor” section, and submit your "Accept" recommendation.

Reviewer #1: All comments have been addressed

Reviewer #2: All comments have been addressed

2. Is the manuscript technically sound, and do the data support the conclusions?

Reviewer #1: (No Response)

Reviewer #2: Yes

3. Has the statistical analysis been performed appropriately and rigorously? 

Reviewer #1: (No Response)

Reviewer #2: Yes

4. Have the authors made all data underlying the findings in their manuscript fully available?

Reviewer #1: (No Response)

Reviewer #2: Yes

5. Is the manuscript presented in an intelligible fashion and written in standard English?

Reviewer #1: (No Response)

Reviewer #2: Yes

6. Review Comments to the Author

Reviewer #1: (No Response)

Reviewer #2: The revision has addressed all the issues that I have raised.

However, the authors should still try to explain how the tumours are distributed after being injected into the Rat's liver. Albeit the liver of the rats are small, how this end up in the peritoneum as carcinomatosis is not explained.

7. PLOS authors have the option to publish the peer review history of their article (what does this mean?). If published, this will include your full peer review and any attached files.

Reviewer #1: No

Reviewer #2: No

---

## [Editor Report · Acceptance letter]

14 Nov 2019

PONE-D-19-16594R1 

The impact of short-term machine perfusion on the risk of cancer recurrence after rat liver transplantation with donors after circulatory death 

Dear Dr. Oldani:

I am pleased to inform you that your manuscript has been deemed suitable for publication in PLOS ONE. Congratulations! Your manuscript is now with our production department. 

With kind regards,

on behalf of

Dr. Ezio Laconi 

Academic Editor

PLOS ONE